# Trust, but Verify: Cross-Modality Fusion for HD Map Change Detection

**John Lambert**[1,2] and **James Hays**[1,2]

[1]Argo AI    [2]Georgia Institute of Technology

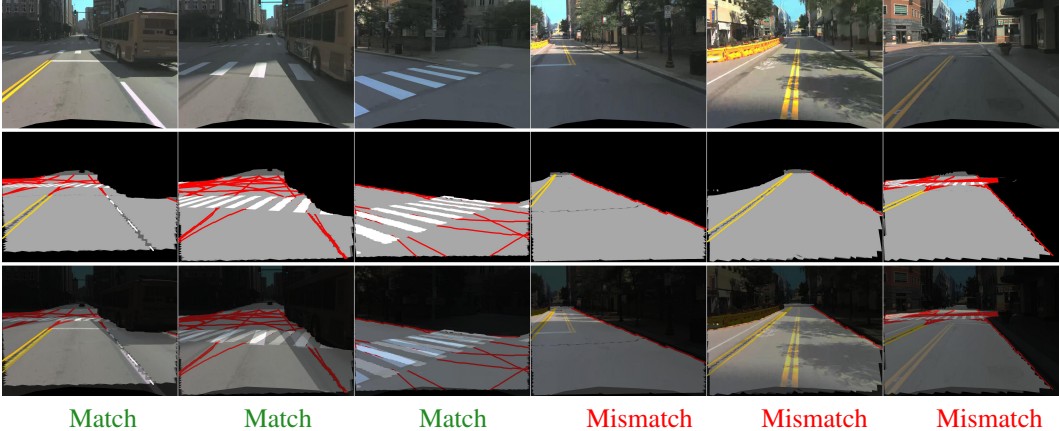

| Match | Match | Match | Mismatch | Mismatch | Mismatch |

Figure 1: Scenes from an example log from our TbV dataset where a real-world change causes an HD map to become stale. Corresponding sensor data (top), map data (middle), and a blended combination of the two (bottom) are shown at 6 timestamps in chronological order from left to right. Within each column, all images are captured or rendered from identical viewpoints. We use red to denote implicit lane boundaries. Faint grey lines in row 1, columns 4-6 show where the previous paint was stripped away.

## Abstract

High-definition (HD) map change detection is the task of determining when sensor data and map data are no longer in agreement with one another due to real-world changes. We collect the first dataset for the task, which we entitle the *Trust, but Verify*[1] (TbV) dataset, by mining thousands of hours of data from over 9 months of autonomous vehicle fleet operations. We present learning-based formulations for solving the problem in the bird's eye view and ego-view. Because real map changes are infrequent and vector maps are easy to synthetically manipulate, we lean on simulated data to train our model. Perhaps surprisingly, we show that such models can generalize to real world distributions. The dataset, consisting of maps and logs collected in six North American cities, is one of the largest AV datasets to date with more than 7.9 million images. We make the data[2] available to the public, along with code and models[3] under the the CC BY-NC-SA 4.0 license.

## 1 Introduction

We live in a highly dynamic world, so much so that significant portions of our environment that we assume to be static are, in fact, in flux. Of particular interest to self-driving vehicle development is changing road infrastructure. Road infrastructure is often represented in an onboard map within a

---

[1]From the Russian proverb, 'Доверяй, но проверяй' ('*Trust, but verify*').

[2]Data is available at Argoverse.org.

[3]Code and models are available at github.com/johnwlambert/tbv.

35th Conference on Neural Information Processing Systems (NeurIPS 2021) Track on Datasets and Benchmarks.

geo-fenced area. Geo-fenced areas have served as an operational design domain for self-driving vehicles since the earliest days of the DARPA Urban Challenge [49, 32, 3].

One way such maps could be used is to constrain navigation in all free space to a set of legal "rails" on which a vehicle can travel. Maps may also be used to assist in planning beyond the sensor range and in harsh environments. Besides providing routes for navigation, maps can ensure that the autonomous vehicle (AV) follows local driving laws when navigating through a city. They embody a representation of the world that the AV can understand, and contain valuable information about the environment. Building and validating maps represent an essential and general part of spatial artificial intelligence (AI) [11], embodied intelligence that enables safety-critical awareness of a robot's surroundings.

However, maps assume a static world, an assumption which is violated in practice; although these changes are rare, they certainly occur and will continue to occur, and can have serious implications. Level 4 autonomy is defined as sustained performance by an autonomous driving system within an operational design domain, without any expectation that a user will respond to a request to intervene [24]. Thus, constant verification that the represented world, expressed as a map, matches the real world, a task known as *map change detection* [35, 34, 26], is a clear requirement for L4 autonomy. This problem may be solved in the *classification* setting, reasoning globally over the map and scene, or additionally in a *localization* setting, where the spatial extent of changed map entities are locally identified. Because dedicated mapping vehicles cannot traverse the world frequently enough to keep maps up to date [34], high-definition (HD) maps become "stale," with out of date information. If maps are used as hard priors, this could lead to confident but incorrect assumptions about the environment.

In this work, we present the first public dataset for urban map change detection based on actual, observed map changes, which we name *TbV*. While researchers could use paired sensor data and HD maps from other datasets such as Argoverse [7] or nuScenes [4] to hypothesize about map change detection performance in the real world based on synthetic data, these datasets include zero real map changes, meaning one could not know with any accuracy or degree of certainty about how well the system would actually operate in the real world. Concurrent work [20] presents qualitative results on a handful of real-world map changes on a proprietary dataset, but relies upon synthetic test datasets for all quantitative evaluation. Not only does no comparable dataset to TbV exist, there also has not even been an attempt to characterize how often map changes occur and what form they take. Collecting data for map change detection is challenging since changes occur randomly and infrequently. In addition, in order to use data corresponding to real changes to train and evaluate models, identified changes must be manually localized in both space and time.

HD map change detection is a difficult task even for humans, as it requires the careful comparison of all nearby semantic entities in the real world with all nearby map elements in the represented world. In an urban scene, there can be dozens of such entities, many with extended shapes. The task is sufficiently difficult that several have even questioned the viability of HD maps for long-term autonomy, opting instead to pursue HD-map-free solutions [26]. We concentrate on changes to two types of semantic entities – lane geometry and pedestrian crosswalks. We define the task as correctly classifying whether a change occurred at evenly spaced intervals along a vehicle's trajectory.

The task itself is relatively new, especially since HD maps were not made publicly available until the release of the Argoverse, nuScenes, Lyft Level5, and Waymo Open Motion datasets [7, 4, 23, 54]. We present the first entirely learning-based formulation for solving the problem in either a bird's eye view (BEV), as well as a new formulation for the ego-view (i.e. front camera frustum), eliminating several heuristics that have defined prior work. We pose the problem as learning a representation of maps, sensor data, or the combination of the two.

Our contributions are as follows:
- We present a novel AV dataset, with 800 vehicle logs in our train and validation splits, and 200 vehicle logs with real-world map-changes in our test split.
- We implement various learning-based approaches as strong baselines to explore this task for the first time with real data. We also demonstrate how gradients flowing through our networks can be leveraged to localize map changes.

- We analyze the advantages of various data viewpoint by training both models operating on the ego-view and others on a bird's eye view.
- We show that synthetic training data is useful for detecting real map changes. At the same time, we identify a considerable domain gap between synthetic and real data, with significant performance consequences.

## 2 Related Work

**HD Maps.** HD maps include lane-level geometry, as well as other geometric data and semantic annotations [31, 30, 52, 55, 21, 7, 22, 59, 15, 14, 36, 28, 9, 47, 46]. The Argoverse [7], nuScenes [4], Lyft Level 5 [23], and Waymo Open Motion [54] datasets are the only publicly available sources of HD maps today, all with different semantic entities. Argoverse [7] includes a ground surface height map, rasterized driveable area, lane centerline geometry, connectivity, and other attributes. nuScenes [4] followed by also releasing centerline geometry, pedestrian crossing polygons, parking areas, and sidewalk polygons, along with rasterized driveable area. Lyft Level 5 [23] later provided a dataset with many more map entities, going beyond lane marking boundaries, crosswalks to provide traffic signs, traffic lights, lane restrictions, and speed bumps. Most recently, the Waymo Open Motion Dataset [54] released motion forecasting scenario data with associated HD maps. Their yet-richer HD map representation includes crosswalk polygons, speed bump polygons, lane boundary polylines with marking type, lane speed limits, lane types, and stop sign positions and their corresponding lane associations; their map data is most comparable with our HD maps. HD maps are useful for a range of tasks, from perception [55, 6, 7], to motion forecasting, to motion planning [8, 59], to traffic scene simulation [47, 46, 9]. All state-of-the-art motion forecasting methods for self-driving today use HD maps [14, 28, 36, 62].

**HD Map Change Detection.** HD map change detection is a recent problem, with limited prior work. Pannen *et al.* [35] introduce one of the first approaches; two particle filters are run simultaneously, with one utilizing only Global Navigation Satellite System (GNSS) and odometry measurements, and the other filter using only odometry with camera lane and road edge detections. These two distributions and sensor innovations are then fed to weak classifiers. Other prior work in the literature seeks to define hand-crafted heuristics for associating online lane and road detections with map entities [25, 45, 34]. These methods are usually evaluated on a single vehicle log [25].

Instead of comparing vector map elements, Ding *et al.* [12] use 2d BEV raster representations of the world; first, IMU-motion-compensated LiDAR odometry is used to build a local online "submap". Afterwards, the submap is projected to 2d and overlaid onto a prebuilt map; the intensity mean, intensity variance, and altitude mean of corresponding cells are compared for change detection. Rather than pursuing this approach, which requires creating and storing high-resolution reflectance maps of a city, we pursue the alignment of *vector maps* with sensor data. Vector maps can be encoded cheaply with low memory cost and are the more common representation, being used in all four public HD map datasets.

In concurrent work, Heo *et al.* [20] introduce an adversarial metric learning-based formulation for HD map change detection, but access to their dataset is restricted to South Korean researchers and performance is measured on a synthetic dataset, rather than on real-world changes. They employ a forward-facing ego-view representation, and require training a second, separate U-Net model to localize changed regions in 2d, whereas we show changed entity localization can come for free via examination of the gradients of a single model.

**Mapping Dynamic Environments.** While "HD maps" are a relatively new entity, dynamic map construction is a more mature field of study. Semi-static environments are not limited to urban streets; households, offices, warehouses, and parking lots are relatively fixed environments that a robot may navigate, with changing cars, furniture, and goods [48]. Mapping dynamic environments has been an area of study within the SLAM community for decades [43, 10, 17]. However, we focus purely on change detection, rather than map updates.

Recently, machine learning for online mapping has generated interest. An alternative to using an HD map prior is to rebuild the map on-the-fly during robot operation; however, such an approach cannot map occluded objects or entities. In addition, these methods are limited to producing raster map layers, such as a driveable area mask, with an output resembling semantic segmentation. Raster data is significantly less useful than vector data for path planning and generating vector map data

Table 1: Probability of a 30m × 30m region that has been visited at least 5 times in 5 months undergoing a lane geometry or crosswalk change within the same time period. These statistics apply only to surface-level urban streets, not highways.

| | CITY NAME | | | | | |
| | PITTSBURGH | DETROIT | WASHINGTON, D.C. | MIAMI | AUSTIN | PALO ALTO |
|---|---|---|---|---|---|---|
| PROBABILITY OF CHANGE | 0.0068 | 0.0056 | 0.0046 | 0.0038 | 0.0009 | 0.0007 |
| Up to T / 1000 TILES WILL CHANGE IN 5 MO. | 7 | 6 | 5 | 4 | 0.9 | 0.7 |

with machine learning is generally an unsolved problem. Raster map layers may be generated from LiDAR [55], accumulated from networks operating on ego-view images over multiple cameras and timesteps [41, 37, 33], or from a single image paired with a depth map or LiDAR [29]. They all show that automatic mapping is quite challenging.

**Image-to-Image Change Detection.** Image-to-image scene change detection over the temporal axis is a well-studied problem [53, 2]. Scenes are dynamic over time in numerous ways, and those ways are mostly nuisance variables for our purposes. We wish to develop models invariant to season, lighting, the fading of road markings, and occlusion because these variables don't actually change the lane geometry. Wang *et al.* [53] introduced the CDnet benchmark, a collection of videos with frame pixels annotated as static, shadow, non-ROI, unknown, or moving. Alcantarilla [2] *et al.* introduce the VL-CMU-CD street view change detection benchmark, from a subset of the Visual Localization CMU dataset.

## 3 The TbV Dataset

We curate a novel dataset of autonomous vehicle driving data comprising 1000 logs, 200 of which contain map changes. The vehicle logs are between 30 and 90 seconds in duration, collected in six North American cities: Austin, TX, Detroit, MI, Miami, FL, Palo Alto, CA, Pittsburgh, PA, and Washington, D.C.

Our training set and validation set consist of real data with accurate corresponding onboard maps ("positives"). Accordingly, synthetic perturbation of positives to create plausible "negatives" is required for training. We release the data, code and API to generate them. However, in the spirit of other datasets meant for testing only (i.e. not training) such as the influential WildDash dataset [58], we curate our test dataset from the real-world distribution. We do so since map changes are difficult to mine [20], thus we save their limited quantity for the test set. We provide a few examples from our 200 test logs in Figure 2. Statistics of the test split are described in the Supplementary Material. We separate 10% of the training data into the held-out validation split.

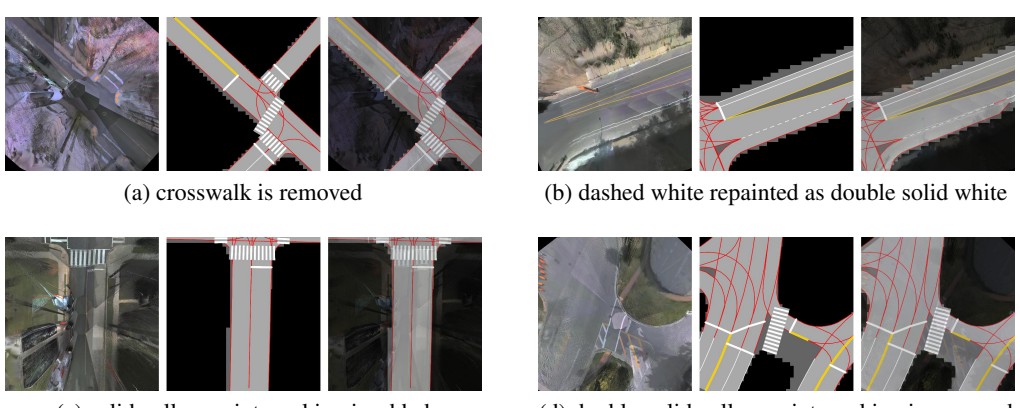

(a) crosswalk is removed  (b) dashed white repainted as double solid white

(c) solid yellow paint marking is added  (d) double-solid yellow paint marking is removed

Figure 2: Examples from the test split of our TbV dataset. Left to right: BEV sensor representation, onboard map representation, blended map and sensor representations. Rows, from top to bottom: deleted crosswalk (top row), and painted lane geometry changes (bottom three rows).

### 3.1 Annotation

In order to label map changes, we use three rounds of human filtering, where changes are identified, confirmed, and characterized by three independent reviewing panels. We assign spatial coordinates to

each changed object within a city coordinate system. Crosswalk changes are denoted by a polygon, and lane geometry changes by polylines. We use egovehicle-to-changed-map-entity distance (point-to-polygon or point-to-line) to determine whether or not a sensor and map rendering should be classified as a map-sensor match or mismatch.

**Analysis of Map Change Frequency** We use our annotated map changes, along with 5 months of fleet data, to analyze the frequency of map changes on a city-scale across several cities. Two particular questions are of interest: (1) *how often will an autonomous vehicle encounter a map change as part of its day-to-day operation?* and (2) *what percentage of map elements in a city will change each month or each year*? For our analysis, we subdivide a city's map into square spatial units of dimension 30 meters × 30 meters, often referred to as "tiles" in the mapping community. We find the probability $p$ of an encounter at any given time with a tile with changed lane geometry or crosswalk to be $p \approx 0.005517\%$. Although the probability of a single event is low, we cannot ignore rare events, as doing so would be reckless. Given the 3.225 trillion miles driven in the U.S. per year [50], this could amount to *billions* of such encounters per year. We determine that up to 7 of every 1000 map tiles may change in a 5-month span (see Table 1), a significant number. More details are provided in the Appendix.

## 3.2 Sensor Data

Our TbV dataset includes LiDAR sweeps collected at 10 Hz, along with 20 fps imagery from 7 cameras positioned to provide a fully panoramic field of view. In addition, camera intrinsics, extrinsics and 6 d.o.f. AV pose in a global coordinate system are provided. LiDAR returns are captured by two 32-beam LiDARs, spinning at 10 Hz in the same direction ("in phase"), but separated in time by $180°$. The cameras trigger in-sync with both of them, leading to a 20 Hz framerate. The 7 global shutter cameras are synchronized to the LiDAR to have their exposure centered on when the LiDAR sweeps through the middle of their fields of view. The top LiDAR spins clockwise in its frame, while the bottom LiDAR spins counter-clockwise in its frame; in the ego-vehicle frame, they both spin clockwise.

## 3.3 Map Data

In the Supplementary Material, we list the semantic map entities we include in the TbV dataset. Previous AV datasets have released sensor data localized within a single map per city [7, 4, 23]. This is not a viable solution for TbV, since the maps change over our long period of data gathering. We instead release local maps with all semantic entities within 20 meters of the egovehicle featured. Accordingly, single, incremental changes can be identified and tested. We release many maps, one per vehicle log; the corresponding map is the map used on-board at time of capture. Lane segments within our map are annotated with boundary annotations for both the right and left marking (including against curbs) and are marked as implicit if there is no corresponding paint.

## 3.4 Dataset Taxonomy

Our dataset's taxonomy is intentionally oriented towards lane geometry and crosswalk changes. In general, we focus on permanent changes, which are far less frequent in urban areas than temporary map changes. Temporary map changes often arise due to construction and road blockades.

We postulate that temporary map changes – temporarily closed lanes or roads, or temporary lanes indicated by barriers or cones, should be relegated to onboard object recognition and detection systems. Indeed, recent datasets such as nuScenes [4] include 3d labeling for traffic cones and movable road barriers, such as Jersey barriers (see Appendix for examples). Even certain types of permanent changes are object-centered (e.g. changes to traffic signs). Accordingly, a natural division arises between "things" and "stuff" in map change detection, just as in general scene understanding [1, 5]. We focus on the "stuff" aspect, corresponding to entities which are often spatially distributed in the BEV; we find lane geometry and crosswalks to be more frequent than other "stuff"-related changes.

# 4 Approach

## 4.1 Learning Formulation

We formulate the learning problem as predicting whether a map is stale by fusing local HD map representations and incoming sensor data. We assume accurate pose is known. At training time, we assume access to training examples in the form of triplets $(x, x^*, y)$, where $x$ is a local region of the

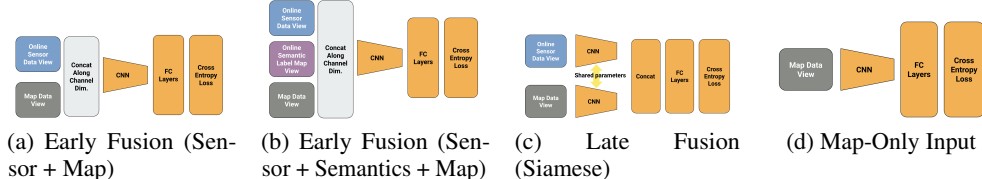

(a) Early Fusion (Sensor + Map) (b) Early Fusion (Sensor + Semantics + Map) (c) Late Fusion (Siamese) (d) Map-Only Input

Figure 3: Learning architectures we explore for the map change detection problem.

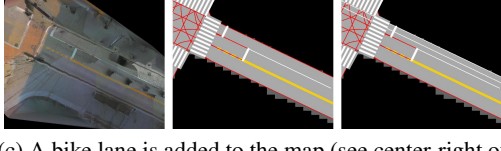

(a) Lane marking color is changed from implicit to solid white (see bottom-center of image)

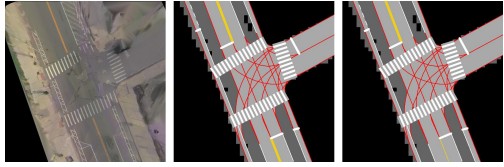

(b) A crosswalk is deleted from the map. Reflections off of windows create illumination variation on the road surface.

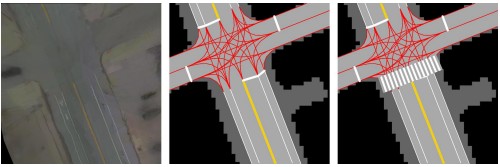

(c) A bike lane is added to the map (see center-right of image)

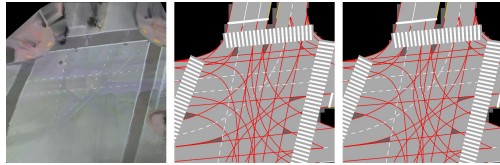

(d) The structure of a lane boundary marking is changed, from double-solid yellow to single-solid yellow (see bottom-center of image). Its color is preserved.

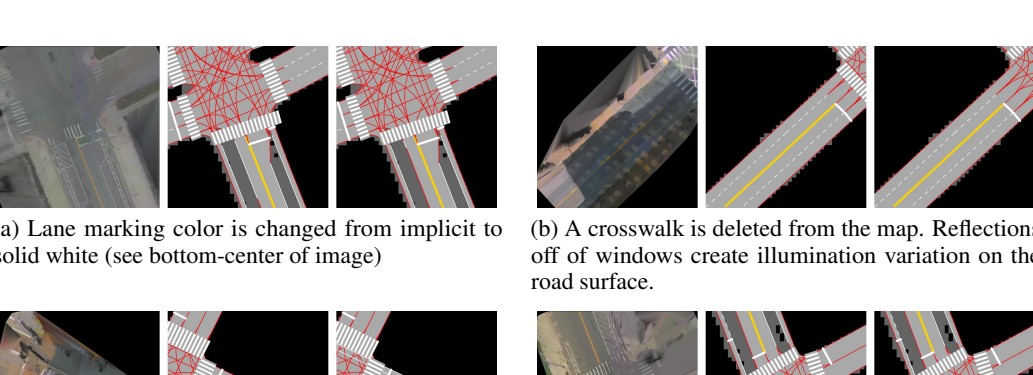

(e) A crosswalk is synthetically inserted into the map.

(f) A solid white lane boundary marking is deleted (see top-center of image).

Figure 4: Examples of our 6 types of synthetic map changes (zoom in for detail). Each row represents a single scene. Left: bird's eye view (BEV) sensor data representation. Center: rasterized onboard map representation (positive). Right: synthetic perturbation of onboard map (negative). We use red to denote implicit lane boundaries.

map, $x^*$ is an online sensor sweep, and $y$ is a binary label suggesting whether a "significant" map change occurred. $(x, x^*)$ should be captured in the same location.

We explore a number of architectures to learn a shared map-sensor representation, including early fusion and late fusion (see Figure 3). The late fusion model uses a siamese network architecture with two input towers, and then a sequence of fully connected layers. We utilize a two-stream architecture [18, 57] with shared parameters, which has been shown to still be effective even for multi-modal input [27]. We also explore an early-fusion architecture, where the map, sensor, and/or semantic segmentation data are immediately concatenated along the channel dimension before being fed to the network. We take no credit for these convnet architectures, which are well studied.

## 4.2 Synthesis of Mismatched Data

Real negatives are difficult to obtain; because their location is difficult to predict a priori, they cannot be captured in a deterministic way by driving around an urban area on any particular day. Therefore, rather than using real negatives for training, we synthesize fake negatives. While sensor data is difficult to simulate, requiring synthesis of sensor measurements from the natural image manifold [56, 9], manipulating vector maps is relatively straightforward.

Table 2: Controlled evaluation of the influence of fusion architecture and scene rendering viewpoint (ego-view vs. BEV). Rows marked with an asterisk represent an expected mean accuracy based on randomly flipped labels, rather than results from a trained model.

| BACKBONE | ARCH. | VIEWPOINT | MODALITIES | | | VISIBILITY-BASED EVAL. @ 20M VAL mACC | BEV PROXIMITY EVAL. @20M | | | VISIBILITY-BASED EVAL. @20M | | |
| | | | RGB | SEMANTICS | MAP | | TEST mACC | IS CHANGED ACC | NO CHANGE ACC | TEST mACC | IS CHANGED ACC | NO CHANGE ACC |
|---|---|---|---|---|---|---|---|---|---|---|---|---|
| - | RANDOM CHANCE* | - | - | - | - | 0.5000 | 0.5000 | 0.50 | 0.50 | 0.5000 | 0.50 | 0.50 |
| RESNET-18 | EARLY FUSION | EGO-VIEW | ✓ | ✓ | ✓ | 0.8417 | 0.6724 | 0.57 | 0.77 | 0.7234 | 0.67 | 0.78 |
| RESNET-18 | LATE FUSION | EGO-VIEW | ✓ | | ✓ | 0.8108 | 0.4930 | 0.13 | 0.85 | 0.4956 | 0.13 | 0.86 |
| RESNET-50 | EARLY FUSION | BEV | ✓ | ✓ | ✓ | 0.9130 | 0.6728 | 0.58 | 0.77 | - | - | - |
| RESNET-50 | LATE FUSION | BEV | ✓ | | ✓ | 0.8697 | 0.5761 | 0.43 | 0.72 | - | - | - |

Table 3: Controlled evaluation of the influence of data modalities. Rows marked with an asterisk represent an expected mean accuracy based on randomly flipped labels, rather than results from a trained model.

| BACKBONE | ARCH. | VIEWPOINT | MODALITIES | | | VISIBILITY-BASED EVAL. @ 20M VAL mACC | BEV PROXIMITY EVAL. @20M | | | VISIBILITY-BASED EVAL. @20M | | |
| | | | RGB | SEMANTICS | MAP | | TEST mACC | IS CHANGED ACC | NO CHANGE ACC | TEST mACC | IS CHANGED ACC | NO CHANGE ACC |
|---|---|---|---|---|---|---|---|---|---|---|---|---|
| - | RANDOM CHANCE* | - | - | - | - | 0.5000 | 0.5000 | 0.50 | 0.50 | 0.5000 | 0.50 | 0.50 |
| RESNET-18 | SINGLE MODALITY* | EGO-VIEW | ✓ | | | 0.5000 | 0.5000 | 0.50 | 0.50 | 0.5000 | 0.50 | 0.50 |
| RESNET-18 | SINGLE MODALITY | EGO-VIEW | | | ✓ | 0.8444 | 0.5333 | 0.36 | 0.70 | 0.5431 | 0.38 | 0.71 |
| RESNET-18 | EARLY FUSION | EGO-VIEW | ✓ | | ✓ | 0.8599 | 0.6463 | 0.52 | 0.77 | 0.6824 | 0.60 | 0.77 |
| RESNET-18 | EARLY FUSION | EGO-VIEW | | ✓ | ✓ | 0.8632 | 0.6082 | 0.36 | 0.85 | 0.6363 | 0.42 | 0.85 |
| RESNET-18 | EARLY FUSION | EGO-VIEW | ✓ | ✓ | ✓ | 0.8417 | 0.6724 | 0.57 | 0.77 | 0.7234 | 0.67 | 0.78 |
| RESNET-50 | SINGLE MODALITY* | BEV | ✓ | | | 0.5000 | 0.5000 | 0.50 | 0.50 | - | - | - |
| RESNET-50 | SINGLE MODALITY | BEV | | | ✓ | 0.8900 | 0.5754 | 0.50 | 0.65 | - | - | - |
| RESNET-50 | EARLY FUSION | BEV | ✓ | | ✓ | 0.9007 | 0.6543 | 0.57 | 0.74 | - | - | - |
| RESNET-50 | EARLY FUSION | BEV | | ✓ | ✓ | 0.9153 | 0.6615 | 0.60 | 0.72 | - | - | - |
| RESNET-50 | EARLY FUSION | BEV | ✓ | ✓ | ✓ | 0.9130 | 0.6728 | 0.58 | 0.77 | - | - | - |

Synthetic data generation via randomized rendering pipelines can be highly effective for synthetic-to-real transfer [44]. In order to synthesize fake negatives from true positives, one must be able to trust the fidelity of labeled true positives. In other words, one must trust that for true positive logs, the map is completely accurate for the corresponding sensor data. We perturb the data in a number of ways (See Supplementary Material). If such fidelity is assured, vector map manipulation is trivial because map elements are vector entities which can be perturbed, deleted, or added.

While synthesizing random vector elements is trivial, sampling from a realistic distribution requires conformance to priors, including the lane graph, drivable area, and intersection. We aim for synthetic map/sensor deviations to resemble real world deviations, and real world deviations tend to be subtle, e.g. a single lane is removed or painted a different color, or a single crosswalk is added, while 90% of the scene is still a match. In order to generate realistic-appearing synthetic map objects, we hand-design a number of priors that must be respected for a perturbed example to enter our training set as a valid training example (see Appendix). Figure 4 and Table 5 of the Supplementary Material enumerate a full list of the 6 types of synthetic changes we employ.

### 4.3 Sensor Data Representation

We experiment with two sensor data representations – *ego-view* (the front center camera image) and *bird's eye view* (BEV). Rather than using Inverse Perspective Mapping (IPM) [16, 60, 40], we generate the BEV representation (i.e. orthoimagery) by ray-casting image pixels to a ground surface triangle mesh. For ray-casting, we use a set of camera sensors with a panoramic field of view, mounted onboard an autonomous vehicle. The temporal aspect is exploited as pixel values from 70 ego-view images are aggregated to render each BEV image (10 timesteps from 7 frustums) in order to reduce sparsity (see Appendix).

### 4.4 Map Data Learning Representation

We render our map inputs as rasterized images; Entities are layered from the back of the raster to the front in the following order: driveable area, lane segment polygons, lane boundaries, pedestrian crossings (i.e. crosswalks). We release the API to generate and render these map images. Vector map entities are synthetically perturbed before rasterization.

## 5 Experimental Results

We frame the map change detection task as: *given a buffer of all past sensor data at timestamp* $t$, *including camera intrinsics and extrinsics, 6 d.o.f. egovehicle pose* $^{city}T_{egovehicle}$ which we denote as $T_{i=0...t}$, *image data* $I^c_{i=0...t}$ *where c is a camera index, lidar sweeps* $L_{i=0...t}$, *onboard map data* $M_{k=0...K}$, *estimate whether the map is in agreement with the sensor data.*

Table 4: Controlled evaluation of the benefit and influence of dropout of data modalities. Rows marked with an asterisk represent an expected mean accuracy based on randomly flipped labels, rather than results from a trained model.

| | | | Modalities | | | Visibility-based Eval @ 20m | BEV Proximity Eval. @20m | | | Visibility-based Eval @20m | | |
| Backbone | Arch. | Viewpoint | RGB | Semantics | Map | Val mAcc | Test mAcc | Is Changed Acc | No Change Acc | Test mAcc | Is Changed Acc | No Change Acc |
|---|---|---|---|---|---|---|---|---|---|---|---|---|
| - | Random Chance* | - | - | - | - | 0.5000 | 0.5000 | 0.50 | 0.50 | 0.5000 | 0.50 | 0.50 |
| ResNet-18 | Early Fusion | Ego-View | ✓ | ✓ | ✓ | 0.8417 | 0.6724 | 0.57 | 0.77 | 0.7234 | 0.67 | 0.78 |
| ResNet-18 | Early Fusion | Ego-View | dropout | dropout | ✓ | **0.8605** | 0.6581 | 0.51 | 0.81 | 0.6926 | 0.58 | **0.81** |
| ResNet-18 | Early Fusion | Ego-View | ✓ | dropout | dropout | 0.8384 | **0.6850** | **0.63** | 0.74 | **0.7342** | **0.72** | 0.74 |
| ResNet-18 | Early Fusion | Ego-View | ✓ | dropout | ✓ | 0.8474 | 0.6483 | 0.51 | 0.78 | 0.6914 | 0.6 | 0.79 |
| ResNet-18 | Early Fusion | Ego-View | ✓ | ✓ | dropout | 0.8429 | 0.6617 | 0.51 | **0.82** | 0.6994 | 0.58 | **0.81** |

## 5.1  Implementation Details

**Ego-view Models.**   Our ego-view models that operate on front-center camera images leverage both LiDAR and RGB sensor imagery. We use LiDAR information to filter out occluded map elements from the rendering. We linearly interpolate a dense depth map from sparse LiDAR measurements, and then compare the depth of individual map elements against the interpolated depth map; elements found behind the depth map are not rendered. In our early fusion architecture, we experiment with models that also have access to semantic label masks from the semantic head of a publicly-available *seamseg* ResNet-50 panoptic segmentation model [39]. For those models with access to the semantic label map modality, we append $224 \times 224$ binary masks for 5 semantic classes ('road', 'bike- lane', 'marking-crosswalk-zebra', 'lane-marking-general', and 'crosswalk-plain') as additional channels in early fusion.

**Bird's Eye View Models.**   We also implement camera-based models that accept orthoimagery as input, relying only upon RGB imagery and a pre-generated ground height field, without utilizing online LiDAR. We generate new orthoimagery each time the ego-vehicle moves at least 5 meters, and use each orthoimagery example as a training or test example. We use 2 cm per pixel resolution for orthoimagery; all pixels corresponding to 3d points up to 20 meters in $\ell_\infty$-norm from the ego-vehicle are included, generating a $2000 \times 2000$ px image.

**Training.**   We use a ResNet-18 or ResNet-50 [19] backbone, with ImageNet-pretrained weights, where a corresponding weight parameter's size is applicable. We use a crop size of $224 \times 224$ from images resized to $234 \times 234$ px. Please refer to the Appendix for additional implementation details and an ablation experiment on the influence of input crop size on performance.

## 5.2  Evaluation

Comparable evaluation of the ego-view and bird's eye view models is challenging since they operate on different portions of the scene. The ego-view model should not be penalized for ignoring changes outside of its field of view, especially those located behind the ego-vehicle. Thus, we provide results for visibility-based evaluation (when the change is visible in the ego-view), and a purely proximity-based comparison (when it is within 20 m. by $\ell_\infty$ norm). The area about which a model should reason is somewhat arbitrary; changes behind and to the side may matter for fleet operation, but changes directly ahead of the AV are arguably most important for path-planning [38]. In addition, changes visible to the rear at some timestamp are often visible directly in front of the AV at a prior timestamp. We consider the visibility-based evaluation to be most fair for ego-view models.

We use a mean of per-class accuracies to measure performance on a two-class problem: predicting whether the real world is changed (i.e. map and sensor data are mismatched), or unchanged (i.e. a valid match). This accounts for both precision and recall. If a confusion matrix is computed with predicted entries on the rows and actual classes as the columns, and normalized by dividing by the sum of each column, 2-class accuracy can be simply calculated as the mean of the diagonal of the confusion matrix. More formally, let $n_{cl} = 2$ be the number of classes, $\hat{y}_i$ be the prediction for the $i$'th test example, and $y_i$ be the ground truth label for the $i$'th test example. We define per-class accuracy ($Acc_c$) and mean accuracy (mAcc) as:

$$\text{mAcc} = \nicefrac{1}{n_{cl}} \sum_{c=0}^{n_{cl}} Acc_c, \quad Acc_c = \frac{\sum_{i=0}^{N} \mathbb{1}\{\hat{y}_i = y_i\} \cdot \mathbb{1}\{y_i = c\}}{\sum_{i=0}^{N} \mathbb{1}\{y_i = c\}} \tag{1}$$

**Which data viewpoint is most effective?**   Models that operate on an ego-view scene perspective are more effective than those operating in the bird's eye view (5% more effective over their own respective field of view), achieving 72.3% mAcc (see Table 2). We found a simpler architecture (ResNet-18) to outperform ResNet-50 in the ego-view.

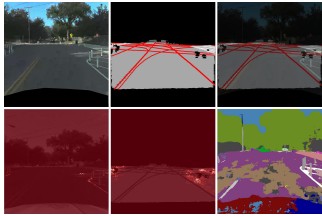
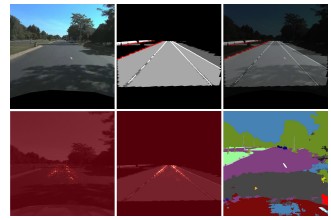
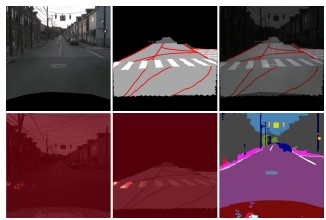

(a) New paint constricting the intersection and bollards are added.

(b) A 3-lane road has been converted to a 2-lane road.

(c) Crosswalk paint has been removed.

Figure 5: Guided GradCAM. 6 figures are shown for frames from various test set logs. Clockwise, from top-left: ego-view sensor image, rendered map in ego-view, blended combination of sensor and map, *seamseg* label map, GradCAM activations for the map input, GradCAM activations for the sensor input. White color show maximal activation, and red color shows zero activation in the heatmap palette. Label maps from *seamseg* are at times quite noisy.

**Which modality fusion method is most effective?**   Early fusion. For both BEV and ego-view, the early fusion models significantly outperform the late fusion models (+22.8% mAcc in the ego-view and +9.7% mAcc in BEV). This may be surprising, but we attribute this to the benefit of early alignment of map and sensor image channels for the early fusion models. Instead, the late-fusion model performs alignment with greatly reduced spatial resolution in a higher-dimensional space, and is forced to make decisions about both data streams *independently*, which may be suboptimal. While the map and sensor images represent different modalities, a shared feature extractor is useful for both.

**Which input modalities are most effective?**   A combination of RGB sensor data, semantics, and the map. We compare validation and test set performance of various input modalities in Table 3. Early fusion of map and sensor data is compared with models that have access to only sensor or map data, or a combination of the two, with or without semantics. All models suffer a significant performance drop on the test set compared to the validation set. While a gap between validation and test performance is undesirable, better synthesis heuristics and better machine learning models can close that gap. We find semantic segmentation label map input to be quite helpful, although it places a dependency upon a separate model at inference time, increasing latency for a real-time system. Mean accuracy improves by 4% in the ego-view and 2% in the BEV when sensor and map data is augmented with semantic information in early fusion. In fact, early fusion of the map with the semantic stream alone (without sensor data) is 1% more effective than using corresponding sensor data for the BEV.

**Is sensor data necessary?**   Yes. The "map-only" model we trained couldn't meaningfully identify map changes over random chance, achieving mean accuracies on the test set of just 0.5754 in the BEV and 0.533 in an ego-view (see Table 3). This model can also be thought of as a binary classifier which is trained to identify whether the HD map is real or synthetically manipulated (i.e. classify the source domain). Inspection via Guided GradCAM demonstrates that the map-only baseline attends to onboard map areas that are not in compliance with real-world priors, such as identifying asymmetry in crosswalk layout, paint patterns, and lane subdivisions.

**Ablation on Modality Dropout.**   We find random drop-out of certain combinations of modalities to regularize the training in a beneficial way, improving accuracy by more than 1% of our best model (See Table 4). Given the wide array of modalities available to solve the task, from RGB sensor data, semantic label maps, rendered maps, and LiDAR reflectance data, we experiment with methods to force the network to learn to extract useful information from multiple modalities. Specifically, we perform random dropout of modalities, an approach developed in the self-supervised learning literature [51, 13, 61].

Perhaps the most intuitive approach would be to apply modality dropout to one of the sensor or semantic streams, forcing the network to extract useful features from both modalities during training. However, we find this is in fact detrimental. More effective, we discover, is to randomly drop out either the map or semantic streams. In theory, meaningful learning should be impossible without access to the map; however, since we drop-out each example in a batch with 50% probability, in expectation 50% of the examples should yield useful gradients in each batch. This approach improves accuracy by more than 1% of our best model. We zero out all data from a specific modality as our drop out technique.

Table 5: Analysis of computational runtime for a single feedforward pass through the network (in milliseconds), for our various map change detection models.

| MODEL (INPUT MODALITIES) | # INPUT CHANNELS | RUNTIME (MS) | |
| --- | --- | --- | --- |
| | | RESNET-18 BACKBONE | RESNET-50 BACKBONE |
| NO FUSION (MAP-ONLY) | 3 | 2.09 | 6.48 |
| NO FUSION (SEMANTIC LABELMAP ONLY) | 5 | 2.10 | 6.64 |
| EARLY FUSION (SENSOR + MAP) | 3 + 3 = 6 | 2.19 | 6.51 |
| EARLY FUSION (SENSOR + MAP + SEMANTICS) | 3 + 3 + 5 = 11 | 2.24 | 6.61 |
| LATE FUSION (SENSOR + MAP) | 3, 3 | 4.29 | 12.99 |

**Interpretability and Localizing Changes.**   While accurately perceiving changes is important, the ability to localize them would also be helpful. Bounding boxes are often unsuitable for a compact localization of a map change because most changed map regions in TbV relate to "stuff" classes, for example, linear, extended lane boundary markings. Instead, pixelwise localization is often more appropriate. We use Guided Grad-CAM [42] to identify which regions of the sensor, map, and semantic input are most relevant to the prediction of the 'is-changed' class. In Figure 5, we show qualitative results on frames for which our best model predicts real-world map changes have occurred.

**Computational Runtime.**   In order to demonstrate that our models can be used in practice without introducing a heavy computational burden, we report the time to complete a feedforward pass through a network, for our various models (see Table 5). The network architectures we employ are computationally lightweight, using ResNet-18 or ResNet-50 backbones. We report feedforward runtime (in milliseconds) for each network, averaged over 100 forward passes, after a warm-up period of 20 forward passes. The hardware used for the analysis is a Quadro P5000 GPU and Intel(R) Xeon(R) W-2145 CPU @ 3.70GHz processor, running the Ubuntu 20.04 operating system. Our best performing map change model (operating in the ego-view) requires just 2.24 milliseconds to complete a forward pass. However, this model does assume access to a readily-available semantic labelmap produced by a semantic segmentation network, which would increase the system latency. In Table 3, we show that for a 4% drop in accuracy for the ego-view models, the semantic labelmap can be excluded from the inputs, in which case the total runtime is just 2.19 milliseconds, well within real-time performance, introducing minimal latency for perception or planning modules which might utilize this information.

# 6   Conclusion

**Discussion.** In this work, we have introduced the first dataset for a new and challenging problem, map change detection. Our dataset is one of the largest AV datasets at the present time, with 1000 logs of duration 30-90 seconds. We implement various approaches as strong baselines to explore this task for the first time with real data. Perhaps surprisingly, we find that comparing maps in a metric representation (a bird's eye view) is inferior to operating directly in the ego-view. We attribute this to a loss of texture during the projection process, and to a more difficult task of reasoning about a much larger spatial area ($85°$ f.o.v. instead of $360°$ f.o.v.). In addition, we provide a new method for localizing changed map entities, thereby facilitating efficient updates to HD maps.

We identify a significant gap between validation accuracy and test accuracy – 10-20% less on the test split – which supports the importance of testing on real data. If performance is only measured on fake changes that resemble one's training distribution, performance can appear much better than what occurs in reality. Real changes can be subtle, and we hope the community will use this dataset to further push the state-of-the-art we introduce. We make publicly available our data, models, and code to generate our dataset and reproduce our results.

**Limitations.    Rendering time.** A second limitation of our work is that real-time rendering requires GPU hardware; in the ego-view, map entity tesselation and rasterization are costly, whereas in the BEV, ray-casting is computationally intensive. **Perturbation diversity.** In our work, we introduce just 6 types of possible map perturbations, of which far more types are possible; nonetheless, we prove that they are surprisingly useful. **Accuracy.** Perhaps last of all, although our baselines have reasonable performance and by inspection we demonstrate they are learning to attend to meaningful regions, a large gap still exists before such a model would be accurate enough to be used on-vehicle.

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
