# OpenReview forum: "Trust, but Verify: Cross-Modality Fusion for HD Map Change Detection"
_NeurIPS.cc/2021/Track/Datasets_and_Benchmarks/Round2 — NeurIPS 2021 Datasets and Benchmarks Track (Round 2)_

### Official Review · Reviewer_fmKM · 2021-09-18
**hd map change detection**

**Rating:** 5
**Confidence:** 2
**Correctness:** The content appears to be correct.

**Strengths:**

1. First dataset to address hd map change detection problem, which is crucial for self-driving system when adopting hd map as a prior.
2. Dataset statistics and annotation pipeline is provided.
3. Experiment results of learning-based methods and detailed ablation studies are presented.
4. Code is available for reproducing the baseline result and dataset can be easily downloaded.

**Weaknesses:**

1. Is there any legal concern about realeasing high-definition map?
2. If government departments and car company cooperate closely and update every road change simultaneously, is there still any significance about this task?
3. I don't really know about this topic, but wonder how many car companies use hd maps as hard priors?
4. As you said in abstract, "Because real map changes are infrequent and vector maps are easy to synthetically manipulate,we lean on simulated data to train our model.". Does this indicate we can use any dataset with hd map to formulate this hd map detection task?


**Additional Feedback:**

NaN

**Clarity:**

The paper is clearly written, with detailed descriptions of the datasets, annotation, and benchmark results.

**Documentation:**

The webpage of this dataset and codebase provide sufficient instructions on how to download and make use of the dataset.

**Ethics:**

No ethical concern

**Relation To Prior Work:**

Discussion with related works is sufficient and clear.

**Summary And Contributions:**

Detecting high-definition (HD) map change is a clear requirement for L4 autonomy due to the reason that confident but incorrect assumptions about the environment will be made with an incorrect hd map. To address this issue, this paper first introduces a hd map change detection dataset called TbV. Based on this dataset, learning-based methods are presented and detailed experiment results and ablation studies are provided.

---

> ### Author Response · Authors · 2021-09-25
> **Thanks for your comments**
>
> Thank you for your time and effort spent in carefully reviewing our work. We are grateful for the encouraging comments ("detailed experiment results", "discussion with related works is sufficient and clear"). Please find our responses to your questions below:
>
> **Q1: Is there any legal concern about releasing high-definition map?**
> In a few countries, this is a concern, such as in South Korea (see lines 104-106 of our paper). However, in many countries, releasing HD maps is perfectly legal, and we have already publicly released HD maps multiple times in the past for the Argoverse Datasets (see [here](https://www.argoverse.org/data.html#download-link)). Other recent datasets have also released HD maps (see lines 77-88 of the main text), such as Lyft L5, [nuScenes](https://github.com/nutonomy/nuscenes-devkit#map-expansion), and the [Waymo Open Motion Dataset](https://blog.waymo.com/2021/03/expanding-waymo-open-dataset-with-interactive-scenario-data-and-new-challenges.html). Argo AI has allowed us to distribute a local HD map per each Trust but Verify (TbV) scenario under a CC BY-NC-SA 4.0 license.
>
> **Q2: If government departments and car company cooperate closely and update every road change simultaneously, is there still any significance about this task?**
> This is a good point. Certainly this cooperation would be desirable, and we would want to see a central, up-to-date repository for each city capable of distributing this information. Digital transmission of real time updates to vehicles from a central city repository is a digital advance almost all cities are unprepared to make at the present time, making map change detection a requirement for self-driving today. In the future, it’s possible that city-to-vehicle over-the-air collaboration may be improved, but this is not a reality today.
>
> Several factors complicate this government-to-company cooperation, such as the use of cascading layers of subcontractors to complete projects by governments. In practice, construction crews do not perform an update at a specified hour of a specified day; rather, roadwork is often completed within a range of some number of weeks, and in some cases, city governments themselves are often not even aware of when roadwork has been completed or taken place. Construction crews do not immediately relay a signal to a citywide traffic network about their work, nor are many crews capable of providing a precise 3d model indicating the localization of the change over the air. Consequently, self-driving cars cannot rely on such information to maintain a constantly up-to-date map.
>
> In short, ultimately trusting or relying on a city government for perfectly accurate information is not a reliable strategy at this time, as the city may be unaware of many factors, and each self-driving vehicle must develop capabilities to be able to trust the road as it actually exists around itself.
>
> **Q4: As you said in abstract, "Because real map changes are infrequent and vector maps are easy to synthetically manipulate, we lean on simulated data to train our model." Does this indicate we can use any dataset with hd map to formulate this hd map detection task?**
> While we could train a model with any collection of accurate HD maps, some of our experiments are not possible to complete using other datasets. For example, no other dataset provides a high-resolution ground surface height map, which is helpful for creating high-quality bird's eye view images from sensor data, enabling our experiments on the value of scene viewpoint.
>
> Most importantly, no other public dataset has ever collected real map changes (not even one such scenario), meaning that while researchers could *hypothesize* about performance in the real world based on synthetic data, they could never know with any accuracy or degree of certainty about how it *actually* works in the real world. Our dataset is the first to include real map change scenarios, and we mine over 200 of them.

---

> > ### Author Response · Authors · 2021-09-25
> > **Using HD maps in industry**
> >
> > **Q3: I don't really know about this topic, but wonder how many car companies use hd maps as hard priors?**
> > This is a good question. Indeed, it is difficult to know exactly how many self-driving companies implement a reliance upon HD maps in their software stack. However, many clues are available in the job descriptions posted by companies today, as well as in companies press releases. Many (if not most) leading companies are using HD maps today such as Waymo, Zoox, NVIDIA, Argo, Motional, TUSimple:
> >
> > ## Motional
> > Motional is currently looking to hire at least 6 engineers to build and maintain their HD maps, and perform HD map change detection:
> > - **Actively Hiring**: [Senior Backend Engineer, Semantic Maps](https://boards.greenhouse.io/motional/jobs/4670512003): “Our services include point cloud labelling, automated map annotations, and map change detection... Work with internal teams focusing on HD mapping applications”
> > - **Actively Hiring**: [Senior Mapping Engineer](https://boards.greenhouse.io/motional/jobs/4653571003): “5+ years of experience in HD Mapping, Localization, State Estimation”
> > - **Actively Hiring**: [Senior Cloud Software Engineer, HD Maps](https://boards.greenhouse.io/motional/jobs/4721848003): “HD Maps enable our autonomous cars to navigate from point A to point B. It is a critical set of information to make the smartest safe decisions on the road.”
> > - **Actively Hiring**: [Mapping Specialist](https://boards.greenhouse.io/motional/jobs/4700070003): “Annotate/digitize high-definition maps of roads that are used for autonomous driving, using in-house mapping software, and play an active role in the expansion and maintenance of mapped areas”
> > - **Actively Hiring**: [Senior Cloud Software Engineer, HD Maps](https://boards.greenhouse.io/motional/jobs/4721848003): “Prior experience with HD Maps and/or mapping for self-driving cars”
> > - **Actively Hiring**: [Map Creator, Front-End](https://boards.greenhouse.io/motional/jobs/4713055003): “Design, plan, and implement future improvements to our specialized, Web-based, graphical semantic map annotation application.”
> >
> > ## Waymo:
> > - Waymo's motion forecasting network named VectorNet, deployed in their fleet, relies upon accurate HD maps (see [Waymo's blog post](https://blog.waymo.com/2020/05/vectornet.html)).
> > - [Waymo's official blog](https://blog.waymo.com/2020/09/the-waymo-driver-handbook-mapping.html) states that “These maps give the Waymo Driver a deep understanding of its environment, from road types and the distance and dimensions of the road itself, to other features like lane merges, stop signs, crosswalks, and so much more. This prior knowledge of what each mile ahead looks like”. Waymo has implemented a type of system for HD map change detection, as described here: “Our system can detect when a road has changed by cross-referencing the real-time sensor data with its on-board map.”
> > - [In their blog](https://blog.waymo.com/2019/09/building-maps-for-self-driving-car.html), Waymo indicates that they rely upon very detailed maps, that include details such as "the height of a curb, width of an intersection, and the exact location of a traffic light or stop sign".
> > - **Actively Hiring**: [ML Engineer, Mapping](https://g.co/kgs/UuAstS), "Improving our internal tools for viewing and editing maps”
> > - **Actively Hiring**: [Software Engineer, Mapping](https://g.co/kgs/ivMkg6)
> >
> > ## Zoox:
> > - Zoox’s publicly-released demo videos [here](https://youtu.be/JAHva2-x1wg?t=2) and [here](https://youtu.be/735PLhalufY?t=55) demonstrate that they have a previously-built HD map that they use in their vehicle.
> > - **Actively Hiring**: [Software Engineer - HD Mapping Algorithms](https://zoox.com/careers/job-opportunity/?job=b78e8490-8fc6-489b-b179-a6875fb2aa94): “you will collaborate in building a highly automated, scalable, multi-modality, 3D mapping system.”
> > - **Actively Hiring**: [Software Engineer - HD Mapping Backend](https://zoox.com/careers/job-opportunity/?job=ae71f479-377b-4260-85ab-0b396990fa25) “You’ll further automate the scalable backend that produces our large-scale 3D maps. The architecture you build will enable us to easily produce new high-fidelity map content at enormous scales in parallel, while delivering rapid updates to small areas where the map is out of date.”
> > - **Actively Hiring**: [Software Engineer - HD Mapping Frontend](https://zoox.com/careers/job-opportunity/?job=e1571b75-ca4e-4795-a1df-88015eafe46d) “design and develop scalable, easy to use tools to enable anyone in the business to understand and interact with our 3D map data”

---

> > > ### Author Response · Authors · 2021-09-25
> > > **Using HD maps in industry**
> > >
> > > ## NVIDIA
> > > - **Actively Hiring**: [Map Operations Engineer](https://nvidia.wd5.myworkdayjobs.com/en-US/NVIDIAExternalCareerSite/job/India-Pune/Map-Operations-Engineer_JR1945756), “Manage the team operating the HD map factory and optimally produce desired mapping mileage on time...Resolve mapping issues reported by Autonomous Vehicles driving using HD maps”
> > > - **Actively Hiring**: [Senior Software Engineer, Mapping - Autonomous Driving](https://nvidia.wd5.myworkdayjobs.com/en-US/NVIDIAExternalCareerSite/job/US-CA-Santa-Clara/Senior-Software-Engineer--Mapping---Autonomous-Driving_JR1945650) “Make sure our algorithm responds well even in the presence of errors in the map, i.e. when the map doesn't correspond with the real world.”
> > > - **Actively Hiring**: [Senior Software Engineer, AV Mapping Infrastructure](https://nvidia.wd5.myworkdayjobs.com/en-US/NVIDIAExternalCareerSite/job/US-CA-Santa-Clara/Senior-Software-Engineer--AV-Mapping-Infrastructure_JR1945841): “Enabling services distributing HD maps for autonomous driving to millions of autonomous vehicles in real-time.”
> > > - **Actively Hiring**: [Frontend Developer – Autonomous Vehicles](https://nvidia.wd5.myworkdayjobs.com/en-US/NVIDIAExternalCareerSite/job/India-Pune/Frontend-Developer---Autonomous-Vehicles_JR1935481) "build, and distribute HD Maps for self-driving cars across the world"
> > > - **Actively Hiring**: [Senior Software Engineer, Mapping - Autonomous Vehicles](https://nvidia.wd5.myworkdayjobs.com/en-US/NVIDIAExternalCareerSite/job/US-WA-Redmond/Senior-Software-Engineer--Mapping---Autonomous-Vehicles_JR1944234) “A detailed HD map is required for localization, it provides critical information for planning, and it increases safety via redundant systems.”
> > > - **Actively Hiring**: [Senior Software Engineer](https://nvidia.wd5.myworkdayjobs.com/en-US/NVIDIAExternalCareerSite/job/US-WA-Redmond/Senior-Software-Engineer_JR1943154) "create and serve HD maps to millions of self-driving cars"
> > > - **Actively Hiring**: [Senior Software Engineer, 3D Graphics, Mapping- Autonomous Vehicles](https://nvidia.wd5.myworkdayjobs.com/en-US/NVIDIAExternalCareerSite/job/US-WA-Redmond/Senior-System-Software-Engineer--3D-Graphics_JR1944222) “Developing scalable cloud native tools to render and edit 3D Maps that can be edited by 100s of labelers.”
> > >
> > > ## TUSimple
> > > - **Actively Hiring**: [Senior Backend Software Engineer, HD Map](https://boards.greenhouse.io/tusimple/jobs/5248016002) “HD Map data is the major offline data on a running autonomous driving vehicle that makes L4 autonomous driving possible...complex map building algorithms and pipelines”
> > > - **Actively Hiring**: [Senior Big Data Infrastructure Engineer, HD Map](https://boards.greenhouse.io/tusimple/jobs/5399264002)
> > > - **Actively Hiring**: [Senior Cloud Infrastructure Engineer, HD Map](https://boards.greenhouse.io/tusimple/jobs/5247919002)
> > > - **Actively Hiring**: [Senior Frontend Software Engineer - Mapping](https://boards.greenhouse.io/tusimple/jobs/5415618002)
> > > - **Actively Hiring**: [Senior Mapping System Engineer - Autonomous Vehicles](https://boards.greenhouse.io/tusimple/jobs/5233537002)
> > >
> > >
> > > ## Toyota Research Institute
> > > - **Actively Hiring**: [Senior or Staff Research Scientist, Data Fusion, Automated Mapping Platform (AMP)](https://jobs.lever.co/tri/07318e2a-22b0-4437-b976-17eebac93417): “The Automated Mapping Platform (AMP) team is responsible for developing a new high definition mapping cloud platform by integrating sensor data from vehicles and global imagery from satellites….The expected output will be an accurate, scalable, semantic HD map of the world.“
> > >
> > > ## Argo AI
> > > Argo AI creates high-definition 3D maps, as discussed in pages 28-29 of their recently released [Safety Report](https://www.argo.ai/wp-content/uploads/2021/04/ArgoSafetyReport.pdf).

---

> > > > ### Comment · Reviewer_fmKM · 2021-09-30
> > > > **Responce to author**
> > > >
> > > > Thanks for your detailed comments about my concerns.
> > > >
> > > > For Q3: I don't really know about this topic, but wonder how many car companies use hd maps as hard priors? Hard priors means that hd maps are regarded as an important part of path planning, and wrong hd maps will have more serious consequences. But if the existing autopilot module mostly relies on vision or point cloud, then I would be a little worried about the necessity of this task. Some job introductions are not enough to judge.
> > > >
> > > > Besides, as you said "Our dataset is the first to include real map change scenarios, and we mine over 200 of them.", I doubt that whether these few numbers of real map change scenarios are enough to evaluate different methods.

---

> > > > > ### Author Response · Authors · 2021-09-30
> > > > > **Use of HD maps and dataset scale**
> > > > >
> > > > > Of course we can't know the details of internal algorithmic details from various autonomous driving efforts (except for Argo AI), so the public statements and public job listings are the best indicator of the importance of HD maps. Our expectation is that HD maps play an important role not just in motion planning, but also motion forecasting, and potentially earlier stages of the perception stack, as well. If HD maps weren't critical, their creation and maintenance would be an expense that could be avoided, and autonomous vehicles could operate more broadly outside of mapped regions.
> > > > >
> > > > > Regardless of the particular algorithmic approaches of self driving companies, map verification is a broader task for robotics, and our dataset is one such benchmark.
> > > > >
> > > > > Note that 200 map change SCENARIOS contain tens of thousands of views of map changes from different vantage points. The 200 map change scenarios are bigger than the entire Argoverse 1.0 tracking dataset. A single scenario might have 3 lanes repainted over a distance of 50 meters, seen in hundreds of frames from multiple camera directions and camera locations. And we show in our paper that these thousands of map change views can be used to evaluate different map change detection architectures and observe clear trends.

---

> > > > > ### Author Response · Authors · 2021-10-01
> > > > > **Response to review fmKM -- discussion about the relationship between the TbV dataset and real-time mapping**
> > > > >
> > > > > Hi Reviewer fmKM, thanks again for your comments.
> > > > >
> > > > > One more thing I think we should note -- for autonomy methods that rely on real-time mapping to gain an understanding of the lanes around them, our TbV dataset is the largest public dataset available for training.

---

### Official Review · Reviewer_5TZF · 2021-09-21
**Better base models and HD map change "detection" (not classification) task**

**Rating:** 6
**Confidence:** 3
**Correctness:** See my comments about detection vs cl…
**Clarity:** The paper is well written.

**Strengths:**

Due to activities in the real world, the layout of the road may be different from the recorded HD map. As a result, map change detection is a critical task. The benchmark proposed in this paper focuses on the recognizing permanent changes. Instead, those temporary changes are left for the on-board object detection and recognition systems. The proposed benchmark provides synthetic training data that (almost) come for free without manual labeling and real-world test data.

**Weaknesses:**

1. There is significant performance drop between the results on synthetic validation data and real-world testing data. Apparently, an immediate research direction is to apply domain adaptation models to this benchmark to bridge the map. But probably results of a simple unsupervised domain adaptation method can be reported in the paper. For example, during training, a binary classifier is trained to tell whether the HD map is real or synthetic.

2. Is it possible to consider the physical constraints in the real-world? For example, if there are no sidewalks on a road, it probably does not make sense to add a crosswalk in the HD map. Of course, it may require some manual effort, but doing so may reduce the domain gap between real-world and synthetic data.

3. Although in the paper, it claims to be focusing on the map change "detection", it is actually a "classification" task. I personally think there is a chance to provide bounding boxes (or other formats) to denote what changes are. But I have to admit I am not an expert in autonomous driving. Not sure if the (real) detection task is an overkill.

**Additional Feedback:**

No.

**Documentation:**

Not clear as the full dataset is not available.

**Ethics:**

No.

**Relation To Prior Work:**

yes.

**Summary And Contributions:**

This paper proposes a benchmark dataset for permanent HD map change detection. The goal is to decide whether the layout of the road (cross walk and lane) changes from the previously captured HD map. Various baseline models are tested on the benchmark and several research problems are identified.

---

> ### Author Response · Authors · 2021-09-26
> **Thanks for your comments**
>
> Thank you for your time and effort spent in carefully reviewing our work. We are grateful for the encouraging comments ("well written"). Please find our responses to your questions below:
>
> **Q1A: There is significant performance drop between the results on synthetic validation data and real-world testing data. Apparently, an immediate research direction is to apply domain adaptation models to this benchmark to bridge the map.**
>
> Thank you for this excellent observation and comment. We agree that there is a considerable amount of possible exploration of domain adaptation in future work using our TbV dataset. The domain gap in TbV between the train/val and test splits arises because of a domain shift between synthetic and real perturbations. The sensor data generally looks identical between the train/val and test splits, although there may be a slightly higher concentration of recently repainted or repaved roads in the test set. Indeed, the difference between the two types of domains is very subtle.
>
> **Q1B: But probably results of a simple unsupervised domain adaptation method can be reported in the paper. For example, during training, a binary classifier is trained to tell whether the HD map is real or synthetic.**
>
> In the main paper, we presented results of a binary classifier that we trained to tell only whether the HD map is real or synthetic (see lines 304-307 of the main text, copied below):
>
> *The map-stream-only models perform slightly better than random chance. Inspection via Guided GradCAM demonstrates that the map-only baseline attends to onboard map areas that are not in compliance with real-world priors, such as symmetry in crosswalk layout, paint patterns, and lane subdivisions.*
>
> The "map-only" model we trained couldn’t meaningfully identify map changes over random chance, achieving mean accuracies on the test set of just 0.5754 in the BEV and 0.533 in an ego-view (see Table 3). We have updated the main text to clarify this.
>
> Could you share with us a few more details about the sort of domain adaptation experiment you would like to see? Are you more interested unsupervised domain adaptation (no labels in target domain) or semi-supervised (limited labels in the target domain)? At training time, all negatives come from the synthetic domain, whereas all positives come from the real domain. We could split the current 200-log test set into 100 validation logs and 100 test logs, and use domain-adaptation based training with some of the real negatives.  Are you interested in the results of a single-source domain (any synthetic change) setting or multiple-source domain (different types of synthetic changes) setting? Is this the type of experiment you would find illuminating?
>
> **Q2: Is it possible to consider the physical constraints in the real-world? For example, if there are no sidewalks on a road, it probably does not make sense to add a crosswalk in the HD map...doing so may reduce the domain gap between real-world and synthetic data.**
>
> Good point. These are reasonable constraints that we could consider adding to our synthetic crosswalk generation process, based on semantic information about sidewalks. We have shown that even with several types of relatively simple synthetic changes, our synthetic training regime can already yield benefits (see Appendix J of the Supplementary Material and Figure 4 of the main text), and we trust that many additional improvements can yet be made. If we consider the problem in an adversarial learning paradigm, then our training data generation scheme can be thought of as an expert-designed "generator" (using knowledge of priors about road network layout), and our trained models as “discriminators". In future work, it might also be interesting to learn the generator directly from data.
>
> **Q3: Although in the paper, it claims to be focusing on the map change "detection", it is actually a "classification" task...a chance to provide bounding boxes (or other formats) to denote what changes are... Not sure if the (real) detection task is an overkill.**
>
> This is also a good point, and we have updated the main text to clarify this. We use the term "map change detection" because it is the currently accepted terminology for the map change classification task (see [Pannen ICRA '19](https://ieeexplore.ieee.org/abstract/document/8794329), [Pannen ICRA '20](https://ieeexplore.ieee.org/document/9197419), [Karpathy CVPRW '20](https://youtu.be/X2CpuabzRaY?t=372) ). Our dataset supports actual 2d or 3d localization of changes for both training and evaluation via the ground truth we provide, and we provided some preliminary results at pixel-wise localization of such changes using Guided GradCAM in lines 323-327 of the main text, as well as in Figures 5a, 5b, and 5c. Since most map changes in TbV relate to "stuff" classes, bounding boxes often cannot compactly describe a changed region (e.g. a single linear extended lane boundary marking).
>
> We look forward to hearing back from you.

---

> > ### Comment · Reviewer_5TZF · 2021-09-29
> > **A simple domain adaptation model**
> >
> > First of all, thank you for addressing all my comments.
> >
> > Regarding the domain adaptation model, I was mainly thinking about an unsupervised model for a single-source main (the simplest model as a baseline). For example, training a binary domain classifier in addition to the map change detection task, like [1][2].
> >
> > Although it may not completely solve the domain different problem, it may show some evidence whether domain adaption is a promising detection for this particular task.
> >
> > Reference
> > [1] Eric Tzeng et al., Adversarial Discriminative Domain Adaptation. CVPR 2017.
> > [2] Yaroslav Ganin and Victor Lempitsky. Unsupervised Domain Adaptation by Backpropagation. ICML 2015.

---

> > > ### Author Response · Authors · 2021-09-30
> > > **Thanks for your comments**
> > >
> > > Thanks for the clarification and for the advice. However, we don't expect this unsupervised domain adaptation method to perform well, for two main reasons which we'll describe below. Nonetheless, we'll work on adding this baseline for the final version of the manuscript.
> > >
> > > The first reason is that the temporal nature of log data makes isolating the specific domain challenging without labels. In other words, our individual data samples ("logs") are not i.i.d. image samples from a single domain, like we might expect to see when comparing examples from the MNIST and SVHN datasets, respectively. Because our domains consist of (1) "real positives" (2) "synthetic negatives," and (3) "real negatives," to isolate out the examples belonging to the "real negative" domain in a temporal sequence (i.e. from a test set sequence), we would need to know at which timestamps a map change is visible. These per-timestamp annotations are equivalent to ground truth "map change" labels (from the target domain). Thus, only a fully supervised domain adaptation method would seem to be reliable. This is because the test logs include 15-90 second sequences where a map change is visible for only a portion of the total log sequence. In other words, half of a test log with a real map change may consist of "real positives", in which there is no domain change, which would lead to unreliable training of the adversarial networks if treated as belonging to the "real negative" target domain, as this was the "real positive" domain seen during training (a source domain).
> > >
> > > The second reason is the following: even if we split off 100 validation logs from the real test split, the 100 unique real map changes contained in them (albeit captured at many separate timestamps, providing many more observations) would be very few in comparison to the hundreds of thousands of unique synthetic examples we generate for training (and likely overwhelmed during training).

---

### Official Review · Reviewer_28PQ · 2021-09-22
**A useful dataset for HD map change detection in autonomous driving.**

**Rating:** 7
**Confidence:** 4
**Correctness:** Good.
**Clarity:** Yes. The paper is well written and we…

**Strengths:**

1. The motivation is clear and the problem is interesting.
2. The dataset collection is non-trivial.
3. The experiments are comprehensive and convicing.
4. The code is open source for the community.

**Weaknesses:**

The two concerns are:
1. Does the task of HD map change detection really play a crucial role in self-driving? What kind of consequences will the HD map change lead to? How the perception and planning modules will be influenced with only a subtle change in HD map, quantitatively or qualitatively? More discussions are required to evaluate the value of studying such a task, since there are many other solutions to promote the robustness of autonomous driving, e.g., online HD map creation. The authors should provide more evidence to show that the task is worth studying.

2. Can the proposed framework be used in practice? The proposed method seems to be quite expensive, yet it should be implemented in real-time for the following-up perception or planning modules which incorporate HD maps. This will further increase the computation burden for the autonomous vehicles. Considering of the low probability of HD map change in reality, should we add this computationally-expensive module to deal with a rare event? Are there any other lightweight and effective solutions?

**Additional Feedback:**

See weaknesses.
***
The authors addressed my concerns, and they also improved the paper with more discussions. So I would like to raise my score from 5 to 7.

**Documentation:**

Yes. The data collection, availability, and maintenance seem good.

**Ethics:**

No.

**Relation To Prior Work:**

Yes. The study of HD map change is scarce and the authors have reviewed existing studies.

**Summary And Contributions:**

This work proposed a nice real-world dataset for HD map change detection, and also studied the cross-modality fusion for the detection performance. Overall, this problem is interesting and the dataset is non-trivial. The paper is well-written and well-organized. The main concerns lie in the significance of the problem and the practical value of the framework.

---

> ### Author Response · Authors · 2021-09-24
> **Thanks for your comments**
>
> Thank you for your time and effort spent in carefully reviewing our work. We are grateful for the encouraging comments (*"problem is interesting", "experiments are comprehensive and convincing"*). Please find our responses to your questions below:
>
> **Question: What kind of consequences will the HD map change lead to?**
> In our [dataset's Github repo](https://github.com/johnwlambert/tbv#dataset-overview), we provide paired before-and-after figures illustrating different types of map changes. Images of these scenarios indicate that if maps are used as hard priors, unsafe behavior is possible. Given the 3.225 trillion miles driven each year in the United States alone, the scale of this problem is very significant (See line 160 of our main paper and this [USDOT Report](https://www.fhwa.dot.gov/pressroom/fhwa1905.cfm)).
>
> **Question: How the perception and planning modules will be influenced with only a subtle change in HD map, quantitatively or qualitatively?** All of the state-of-the-art forecasting methods today use HD maps, so a subtle change in such a map could lead to undesirable behavior (see [TNT, CoRL '20](https://corlconf.github.io/corl2021/paper_189/), [VectorNet, CVPR '20](https://openaccess.thecvf.com/content_CVPR_2020/papers/Gao_VectorNet_Encoding_HD_Maps_and_Agent_Dynamics_From_Vectorized_Representation_CVPR_2020_paper.pdf), [LaneGCN, ECCV '20](https://www.ecva.net/papers/eccv_2020/papers_ECCV/papers/123470528.pdf), [CoverNet, CVPR '20](https://openaccess.thecvf.com/content_CVPR_2020/papers/Phan-Minh_CoverNet_Multimodal_Behavior_Prediction_Using_Trajectory_Sets_CVPR_2020_paper.pdf)).
>
> **Question: Does the task of HD map change detection really play a crucial role in self-driving? ... More discussions are required to evaluate the value of studying such a task, since there are many other solutions to promote the robustness of autonomous driving, e.g., online HD map creation. The authors should provide more evidence to show that the task is worth studying.**
>
> Spatial AI, or embodied intelligence that allows robots to operate autonomously in 3D space, enables safety-critical awareness of a robot's surroundings. Building and validating maps are an essential and general part of spatial artificial intelligence (See Andrew Davison's [FutureMapping, 2018](https://arxiv.org/abs/1803.11288)).
>
> Map change detection, a type of online map validation, is a fairly well-known problem in industry, but less well-known in academia. We hope our new dataset and code release will enable academic researchers to address this problem for the first time.
>
> Indeed, the community has not converged on the best long-term approach to self-driving, but many (if not most) leading companies are using HD maps today such as Waymo, Zoox, NVIDIA, Argo, Motional, TUSimple, as indicated in the following press materials and open job positions for which they are actively hiring:
>
> ## Motional
> Motional is currently looking to hire at least 6 engineers to build and maintain their HD maps, and perform HD map change detection:
> - **Actively Hiring**: [Senior Backend Engineer, Semantic Maps](https://boards.greenhouse.io/motional/jobs/4670512003): “Our services include point cloud labelling, automated map annotations, and map change detection... Work with internal teams focusing on HD mapping applications”
> - **Actively Hiring**: [Senior Mapping Engineer](https://boards.greenhouse.io/motional/jobs/4653571003): “5+ years of experience in HD Mapping, Localization, State Estimation”
> - **Actively Hiring**: [Senior Cloud Software Engineer, HD Maps](https://boards.greenhouse.io/motional/jobs/4721848003): “HD Maps enable our autonomous cars to navigate from point A to point B. It is a critical set of information to make the smartest safe decisions on the road.”
> - **Actively Hiring**: [Mapping Specialist](https://boards.greenhouse.io/motional/jobs/4700070003): “Annotate/digitize high-definition maps of roads that are used for autonomous driving, using in-house mapping software, and play an active role in the expansion and maintenance of mapped areas”
> - **Actively Hiring**: [Senior Cloud Software Engineer, HD Maps](https://boards.greenhouse.io/motional/jobs/4721848003): “Prior experience with HD Maps and/or mapping for self-driving cars”
> - **Actively Hiring**: [Map Creator, Front-End](https://boards.greenhouse.io/motional/jobs/4713055003): “Design, plan, and implement future improvements to our specialized, Web-based, graphical semantic map annotation application.”

---

> > ### Author Response · Authors · 2021-09-24
> > **Value of HD maps and HD map change detection**
> >
> >
> > ## Waymo:
> > - Waymo's motion forecasting network named VectorNet, deployed in their fleet, relies upon accurate HD maps (see [Waymo's blog post](https://blog.waymo.com/2020/05/vectornet.html)).
> > - [Waymo's official blog](https://blog.waymo.com/2020/09/the-waymo-driver-handbook-mapping.html) states that “These maps give the Waymo Driver a deep understanding of its environment, from road types and the distance and dimensions of the road itself, to other features like lane merges, stop signs, crosswalks, and so much more. This prior knowledge of what each mile ahead looks like”. Waymo has implemented a type of system for HD map change detection, as described here: “Our system can detect when a road has changed by cross-referencing the real-time sensor data with its on-board map.”
> > - [In their blog](https://blog.waymo.com/2019/09/building-maps-for-self-driving-car.html), Waymo indicates that they rely upon very detailed maps, that include details such as "the height of a curb, width of an intersection, and the exact location of a traffic light or stop sign".
> > - **Actively Hiring**: [ML Engineer, Mapping](https://g.co/kgs/UuAstS), "Improving our internal tools for viewing and editing maps”
> > - **Actively Hiring**: [Software Engineer, Mapping](https://g.co/kgs/ivMkg6)
> >
> > ## Zoox:
> > - Zoox’s publicly-released demo videos [here](https://youtu.be/JAHva2-x1wg?t=2) and [here](https://youtu.be/735PLhalufY?t=55) demonstrate that they have a previously-built HD map that they use in their vehicle.
> > - **Actively Hiring**: [Software Engineer - HD Mapping Algorithms](https://zoox.com/careers/job-opportunity/?job=b78e8490-8fc6-489b-b179-a6875fb2aa94): “you will collaborate in building a highly automated, scalable, multi-modality, 3D mapping system.”
> > - **Actively Hiring**: [Software Engineer - HD Mapping Backend](https://zoox.com/careers/job-opportunity/?job=ae71f479-377b-4260-85ab-0b396990fa25) “You’ll further automate the scalable backend that produces our large-scale 3D maps. The architecture you build will enable us to easily produce new high-fidelity map content at enormous scales in parallel, while delivering rapid updates to small areas where the map is out of date.”
> > - **Actively Hiring**: [Software Engineer - HD Mapping Frontend](https://zoox.com/careers/job-opportunity/?job=e1571b75-ca4e-4795-a1df-88015eafe46d) “design and develop scalable, easy to use tools to enable anyone in the business to understand and interact with our 3D map data”
> >
> >
> > ## NVIDIA
> > - **Actively Hiring**: [Map Operations Engineer](https://nvidia.wd5.myworkdayjobs.com/en-US/NVIDIAExternalCareerSite/job/India-Pune/Map-Operations-Engineer_JR1945756), “Manage the team operating the HD map factory and optimally produce desired mapping mileage on time...Resolve mapping issues reported by Autonomous Vehicles driving using HD maps”
> > - **Actively Hiring**: [Senior Software Engineer, Mapping - Autonomous Driving](https://nvidia.wd5.myworkdayjobs.com/en-US/NVIDIAExternalCareerSite/job/US-CA-Santa-Clara/Senior-Software-Engineer--Mapping---Autonomous-Driving_JR1945650) “Make sure our algorithm responds well even in the presence of errors in the map, i.e. when the map doesn't correspond with the real world.”
> > - **Actively Hiring**: [Senior Software Engineer, AV Mapping Infrastructure](https://nvidia.wd5.myworkdayjobs.com/en-US/NVIDIAExternalCareerSite/job/US-CA-Santa-Clara/Senior-Software-Engineer--AV-Mapping-Infrastructure_JR1945841): “Enabling services distributing HD maps for autonomous driving to millions of autonomous vehicles in real-time.”
> > - **Actively Hiring**: [Frontend Developer – Autonomous Vehicles](https://nvidia.wd5.myworkdayjobs.com/en-US/NVIDIAExternalCareerSite/job/India-Pune/Frontend-Developer---Autonomous-Vehicles_JR1935481) "build, and distribute HD Maps for self-driving cars across the world"
> > - **Actively Hiring**: [Senior Software Engineer, Mapping - Autonomous Vehicles](https://nvidia.wd5.myworkdayjobs.com/en-US/NVIDIAExternalCareerSite/job/US-WA-Redmond/Senior-Software-Engineer--Mapping---Autonomous-Vehicles_JR1944234) “A detailed HD map is required for localization, it provides critical information for planning, and it increases safety via redundant systems.”
> > - **Actively Hiring**: [Senior Software Engineer](https://nvidia.wd5.myworkdayjobs.com/en-US/NVIDIAExternalCareerSite/job/US-WA-Redmond/Senior-Software-Engineer_JR1943154) "create and serve HD maps to millions of self-driving cars"
> > - **Actively Hiring**: [Senior Software Engineer, 3D Graphics, Mapping- Autonomous Vehicles](https://nvidia.wd5.myworkdayjobs.com/en-US/NVIDIAExternalCareerSite/job/US-WA-Redmond/Senior-System-Software-Engineer--3D-Graphics_JR1944222) “Developing scalable cloud native tools to render and edit 3D Maps that can be edited by 100s of labelers.”

---

> > > ### Author Response · Authors · 2021-09-24
> > > **Value of HD maps and HD map change detection**
> > >
> > > ## TUSimple
> > > - **Actively Hiring**: [Senior Backend Software Engineer, HD Map](https://boards.greenhouse.io/tusimple/jobs/5248016002) “HD Map data is the major offline data on a running autonomous driving vehicle that makes L4 autonomous driving possible...complex map building algorithms and pipelines”
> > > - **Actively Hiring**: [Senior Big Data Infrastructure Engineer, HD Map](https://boards.greenhouse.io/tusimple/jobs/5399264002)
> > > - **Actively Hiring**: [Senior Cloud Infrastructure Engineer, HD Map](https://boards.greenhouse.io/tusimple/jobs/5247919002)
> > > - **Actively Hiring**: [Senior Frontend Software Engineer - Mapping](https://boards.greenhouse.io/tusimple/jobs/5415618002)
> > > - **Actively Hiring**: [Senior Mapping System Engineer - Autonomous Vehicles](https://boards.greenhouse.io/tusimple/jobs/5233537002)
> > >
> > >
> > > ## Toyota Research Institute
> > > - **Actively Hiring**: [Senior or Staff Research Scientist, Data Fusion, Automated Mapping Platform (AMP)](https://jobs.lever.co/tri/07318e2a-22b0-4437-b976-17eebac93417): “The Automated Mapping Platform (AMP) team is responsible for developing a new high definition mapping cloud platform by integrating sensor data from vehicles and global imagery from satellites….The expected output will be an accurate, scalable, semantic HD map of the world.“
> > >
> > > ## Argo AI
> > > Argo AI creates high-definition 3D maps, as discussed in pages 28-28 of their recently released [Safety Report](https://www.argo.ai/wp-content/uploads/2021/04/ArgoSafetyReport.pdf).

---

### Official Review · Reviewer_6Sjx · 2021-09-23
**TbV Review**

**Rating:** 7
**Confidence:** 3
**Correctness:** The dataset was annotated carefully t…
**Clarity:** Yes, the exposition is very clear.

**Strengths:**

The need for the map change detection dataset is strongly motivated.

The authors show that training on synthetic data can lead to decent generalization onto real test data, but with lower accuracy than on synthetic validation data.

The authors' findings include (1) modality fusion is better done early than later, (2) combining RGB, semantics, and map information is most beneficial for generalizing to test data, and (3) comparing maps is better done from an egocentric perspective than from a top-down perspective.

**Weaknesses:**

I believe the formulation of the mean accuracy metric should be moved to the main text.

Aside from that, I found that the main text and appendix addressed all of the questions I had while reading.

**Additional Feedback:**

N/A.

**Documentation:**

The repository for the dataset is documented well.

Instructions for running the code are clear.

**Relation To Prior Work:**

The authors thoroughly establish the differences between their work and prior work.

**Summary And Contributions:**

The authors contribute a substantial dataset and baseline models for addressing map change detection from egocentric and top-down perspectives and using various sources of information.

---

> ### Author Response · Authors · 2021-09-30
> **Thanks for your comments**
>
> Thank you for the encouraging feedback.
>
> **Q1: I believe the formulation of the mean accuracy metric should be moved to the main text.**
>
> We have updated the main text to add this detail.
>
> **Strength 1: The need for the map change detection dataset is strongly motivated.**
>
> We agree, and consider the collection of this new public dataset to be a breakthrough for self-driving mapping research, and more broadly, for opportunities to explore the interaction between learning and (lifelong) mapping. Several individuals who are eager to use and explore the dataset have already contacted us about it.
>
>
> **Strength 3: The authors' findings include (1) modality fusion is better done early than later, (2) combining RGB, semantics, and map information is most beneficial for generalizing to test data, and (3) comparing maps is better done from an egocentric perspective than from a top-down perspective.**
>
> That’s correct. For the sake of discussion, some of our findings could be results that are particular to the representations we use for map and sensor data. The dataset supports people finding new and better representations.

---

### Author Response · Authors · 2021-09-30
**Common response**

We thank the reviewers for the insightful comments and questions. The reviewers appreciate that the map change problem problem is interesting (R-28PQ), represents a critical task (R-5TZF) for self-driving, and recognized the need and strong motivation for our TbV dataset (R-6Sjx, R-28PQ).

The reviewers value the novelty of the dataset, as it represents the first dataset to address the HD map change detection problem (R-fmKM). Reviewers noted that the experiments are comprehensive and convicing (R-28PQ), and the paper and exposition is well written and well organized (R-5TZF, R-28PQ, R-fmKM). The reviewers also appreciated the quality of the dataset annotation (R-6Sjx), and appreciated that the code to reproduce the experiments has been open-sourced for the community and is clearly documented ((R-6Sjx, R-28PQ, R-fmKM).

We respond to specific queries below and incorporate all feedback in the updated, uploaded draft of the paper.

---

### Decision · Program_Chairs · 2021-10-10

**Decision:**

Accept

**Comment:**

3 out of 4 reviewers recommend accepting the paper and the concerns of the 4th reviewer are clearly discussed in the authors' feedback. Overall, HD maps have been heavily used in production systems for autonomous driving. The authors leverage the industrial resources and open the venue for researchers to study the HD map maintenance problem. The contribution can be very important for the machine learning research.